# Effect of Oxytocin, Cloprostenol or Buserelin in Semen Doses on Sow Fertility

**DOI:** 10.3390/ani9100746

**Published:** 2019-09-29

**Authors:** Rodrigo Manjarín, Roy N. Kirkwood, Jose Ngula, Felipe Martinez-Pastor, Beatrix Alegre, Juan Carlos Domínguez

**Affiliations:** 1Animal Science Department, California Polytechnic State University, San Luis Obispo, CA 93407-0255, USA; rmanjari@calpoly.edu; 2School of Animal and Veterinary Sciences, University of Adelaide, 5371 Roseworthy, Australia; 3Department of Medicine, Surgery and Veterinary Anatomy, Universidad de León, s/n, 24071 León, Spain; etcjna00@estudiantes.unileon.es (J.N.); ctibag@unileon.es (B.A.); jcdomt@unileon.es (J.C.D.); 4Molecular Biology (Cell Biology), Universidad de León, s/n, 24071 León, Spain; felipe.martinez@unileon.es; 5Institute of Animal Health and Pastoral Farming Development, Universidad de León, s/n, 24071 León, Spain

**Keywords:** artificial insemination, seminal additives, sows, fertility

## Abstract

**Simple Summary:**

Efficient pork production relies on a consistent supply of market pigs. However, many sows experience a seasonal infertility during the hotter months, resulting in fewer pigs produced and a seasonally constrained pork supply. The present study examined the impact of supplementing boar semen with different hormones in order to combat sow seasonal infertility. The results confirmed a seasonal infertility, in that litter sizes were reduced for sows bred during May to August, but that this adverse effect could be reversed by adding hormones to the semen doses used to inseminate sows.

**Abstract:**

During the periods January to April, May to August, and September to December in two consecutive years, sows were assigned at breeding to receive semen doses supplemented with 87 µg cloprostenol (PG; *n* = 158), 5 IU oxytocin (OT; *n* = 154), 2 µg buserelin (GN; *n* = 93), or served as non-supplemented controls (CON; *n* = 605). Sows were inseminated at the detection of estrus, and again 24 h later, but only the first inseminations were supplemented. Compared to CON, only buserelin increased pregnancy and farrowing rates (*p* ≤ 0.05); there was no effect of a period or a treatment × period interaction. Litter size was larger *(p* ≤ 0.001) for all seminal additive groups during the first two periods and tended to increase in GN compared to CON (*p* ≤ 0.1) during the third period, resulting in a tendency (*p* < 0.1) for a period × treatment interaction. The addition of cloprostenol, oxytocin or buserelin to semen doses at first insemination increases litter size in multiparous sows.

## 1. Introduction

Controlling the costs of production are necessary for farm profitability, and for pork producers, this involves maximizing reproductive output. Artificial insemination (AI) provides labor efficiencies and genetic benefits over natural mating, but can be associated with a decreased reproductive performance if not performed properly. The transport of sperm from the site of deposition to the sperm reservoir requires appropriate myometrial contractility and, if compromised, the subsequent fertilization rates and litter sizes may be reduced [1]. During natural breeding, the presence of the boar induces central oxytocin release in the sow increasing uterine activity [2,3]. In addition, estrogens in boar seminal plasma increase myometrial contractions by causing the local release of prostaglandin (PG) F2α from the endometrium [4]. Further, seminal estrogens can enter the circulation, influencing endogenous gonadotrophin release and time of ovulation [1]. Improved timing of insemination relative to time of ovulation will improve sow fertility [5].

Improvements in farrowing rate and/or litter size have followed the addition to the semen dose of estrogen [6], PGF2α [7] and oxytocin [8,9]. However, the results of semen supplementation have been inconsistent with beneficial effects more evident during periods of seasonal infertility [7,8]. Rabbits are induced ovulators, but a study in rabbits investigating use of the Gonadotropin Releasing Hormone (GnRH) analogs buserelin and lecirelin in seminal doses showed improved ovulation rates compared to controls [10]. We are not aware of any reproductive data for pigs pertaining to the supplementation of semen doses with GnRH analogs. Therefore, the objectives of this study were to re-assess the effect of previously described seminal additives on sow reproductive performance throughout the year, and to compare them to the inclusion of buserelin in seminal doses.

## 2. Materials and Methods

This study was performed on an 800-sow farrow-to-finish facility near Burgos, Spain with the knowledge of the University of León Animal Care Committee. Specific permission was not required as all procedures involved normal commercial management. The farm was located in a continental-Mediterranean climate zone, and sow performance was monitored during three periods: January to April, May to August, and September to December, in two consecutive years. The sow accommodations included cooling panels limiting the maximum temperature to 28 °C. During the 24-month study period, the fertility of 1010 multiparous Landrace × Large White sows were examined in response to different seminal additives. At weaning, sows were housed in individual gestation stalls until confirmed pregnant at 35 d after mating by transabdominal ultrasound. Thereafter, sows were rehoused into pens each housing 8–10 sows providing a minimum of 1.5 m^2^ per sow.

Sows had fenceline boar contact for 5 min/d for up to 15 d from weaning to facilitate detection of the post-weaning estrus. Sows were assigned by parity to insemination of pooled semen doses containing 3 × 10^9^ sperm in 80 mL extender at estrus detection, and 24 h later if still exhibiting estrus behavior. Semen doses were supplemented with either 87 µg of cloprostenol (PG; 1 mL Planate^®^, MSD Animal Health, Madison, NJ, USA; *n* = 158), 5 IU oxytocin (OT; 0.5 mL Ovivex^®^, S.P. Veterinaria, Tarragona, Spain; *n* = 154), 2 µg GnRH analog buserelin (GN; 0.5 mL Receptal^®^, MSD Animal Health; *n* = 93), or were non-supplemented controls (CON; *n* = 605). The higher number of controls was due to the farm limiting risk by restricting the number of sows assigned to treatment, especially sows receiving GN as its effect on production was unknown, with all other sows of appropriate parity in each breeding week being assigned to CON. Additives were included in the seminal dose approximately 15 min prior to the first insemination only, as we have previously shown no additional effect of oestrogen and oxytocin supplementation during a second insemination [11].

The dose of oxytocin was based on its efficacy in previous studies [9,12], whereas that of buserelin was based on published work in rabbits [10]. The cloprostenol dose was chosen based on a previous study assuming that 1.0 µg of cloprostenol had similar activity to 50 µg of PGF2α [7]. Sows went to term to allow the determination of farrowing rates and litter sizes.

Differences in farrowing rates between groups were examined by logistic regression analysis using a generalized linear mixed model in SAS 9.2 (PROC GLIMMIX; SAS Institute Inc., Cary, NC, USA), assuming a binary distribution of the response variables. The linear model included treatment, period and their interaction as fixed effects and insemination group as random effect. Differences in total litter size were analyzed by a two-way ANOVA using a linear mixed model (PROC MIXED) that included the same parameters as above. Normality of the residuals and presence of outliers were assessed by PROC UNIVARIATE using the Shapiro-Wilk test, Q-Q-plot, and externally studentized residuals. Litter size was power transformed by a parameter φ whose optimal value was estimated using the maximum likelihood method [13]. *p*-values for pre-planned pairwise comparisons were calculated using Student’s *t*-tests. Data are presented as probabilities and least square means ± SE, with effects considered significant at *p* ≤ 0.05.

## 3. Results

Pregnancy and farrowing rates increased (*p* ≤ 0.05) in GN compared to CON sows with PG and OT sows being intermediate (Table 1). There were no effects of period (*p* = 0.11 and 0.18) or period × treatment (*p* = 0.81 and 0.75). Regarding litter size, there were effects of treatment (*p* ≤ 0.0001) and period (*p* ≤ 0.05), and a tendency for a treatment × period interaction (*p* = 0.09; Table 1). Compared to CON, litter size was higher *(p* ≤ 0.001) for all seminal additive groups during the first two periods, and tended to increase in GN compared to CON (*p* ≤ 0.1) during the third period with other treatments being intermediate (Table 1).

## 4. Discussion

The results of this study indicate that the seminal additives we used can increase pregnancy and farrowing rates although, with the exception of buserelin, the effect was not significant. The lack of improvement in farrowing rate in response to seminal additives likely reflects the excellent farrowing rates in this herd leaving little or no room for improvement. There was also no effect of period on farrowing rates, which probably results from the controlled temperatures in the farrowing accommodations ameliorating any adverse seasonal effect on nutrient intakes. In contrast, compared to CON, litter size was significantly increased by all additives during periods one and two. In period three, litter size tended to increase in response to buserelin with other treatments being intermediate.

The positive effect of buserelin on reproductive performance suggests a possible role of seminal additives on timing of ovulation. Previous work in rabbits did demonstrate an effect of seminal buserelin on ovulation [14], although the pattern of LH release was of an attenuated LH surge. Given that under conditions of uterine estrogen dominance oxytocin will induce endogenous PGF2α release, and that high preovulatory intrafollicular PGF2α concentrations are a requirement for ovulation [15], it is plausible that seminal oxytocin may also influence timing of ovulation. In support of this, blockade of PGF2α production stops ovulation [16] and the administration of cloprostenol advances time of ovulation in gilts [17].

The lack of significant effects on litter size during the latter part of the year in response to oxytocin and PGF2α is interesting. Although speculative, it is possible that the hormonal threshold levels needed to increase myometrial contractility were higher during May through August, reducing the effectiveness of oxytocin and PGF2α in the semen. However, our farrowing data contrasts with those of previous research that demonstrated higher farrowing rates during July through December associated with supplemental seminal oxytocin [9] or PGF2α [7]. Although speculative in the absence of specific data, the improvements noted by the latter authors are likely due to the lack of environmental control in their farrowing facilities, with attendant impaired feed intake and post-weaning infertility. Under these conditions, the seminal additives would be more likely to be effective. Furthermore, in contrast to the present results, the latter authors further noted increased litter sizes during the second half of the year associated with both oxytocin and PGF2α seminal additives, again possibly reflecting lactation feed intakes. The mechanism involved in the previously described effects of oxytocin or PGF2α presumably involves stimulation of uterine contractions, thus promoting sperm transport towards the uterotubal junction. Under these conditions, a larger sperm reservoir in the oviduct may result, with a positive impact on fertility. Responses to exogenous oxytocin are variable which may involve the degree of pre-existing naturally driven myometrial contractions. Under conditions of normal uterine contractility, the direction of sperm transport is towards the uterotubal junction with positive effects on subsequent fertility; this may also occur in response to oxytocin in sows having otherwise poor uterine contractility. However, if the sow is having normal uterine contractility, the response to supplemental oxytocin may be more uncoordinated uterine contractility, with attendant increased sperm backflow and decreased fertility [18]. 

## 5. Conclusions

Based on the present results, we have demonstrated a role for seminal additives in promoting sow fertility to artificial insemination. The greatest and most consistent effect resulted from the supplementation of seminal doses with buserelin.

## Figures and Tables

**Table 1 animals-09-00746-t001:** Effect of seminal additives on fertility of multiparous sows throughout the year. Sows were assigned by parity to insemination of semen doses containing either 87 µg of cloprostenol (PG), 5 IU oxytocin (OT), 2 µg buserelin (GN), or serve as non-treated controls.

Item ^1^	PG	OT	GN	Control
No. of sows	158	154	93	605
Pregnant, % ^2^	89.2% ^cd^	90.3% ^cd^	94.6% ^c^	88.1% ^d^
Farrowing rate, % ^3^	91.4 ^cd^	89.1 ^cd^	93.2 ^c^	84.9 ^d^
Total born litter size (±SE) ^4^				
January–April	15.3 ^e^ ± 0.53	15.0 ^e^ ± 0.62	15.5 ^e^ ± 0.63	13.4 ^f^ ± 0.32
May–August	15.1 ^c^ ± 0.58	14.6 ^c^ ± 0.52	14.2 ^c^ ± 0.62	12.4 ^d^ ± 0.28
September–December	14.0 ^ab^ ± 0.45	13.7 ^ab^ ± 0.46	14.6 ^a^ ± 0.65	13.4 ^b^ ± 0.27

^1^ Within row, values with different superscripts are different: ^a,b^
*p* ≤ 0.1, ^c,d^
*p* ≤ 0.05, ^e,f^
*p* ≤ 0.001. ^2^ There were no effects of period (*p* = 0.11) or period × treatment (*p* = 0.81). ^3^ There were no effects of period (*p* = 0.18) or period × treatment (*p* = 0.75). ^4^ Effect of treatment (*p* ≤ 0.0001) and period (*p* ≤ 0.05). There was a tendency of treatment × period interaction (*p* = 0.09).

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
