# Peer review of "Effect of Oxytocin, Cloprostenol or Buserelin in Semen Doses on Sow Fertility"

_animals, 2019, doi:10.3390/ani9100746_

Round 1

Reviewer 1 Report

Dear Authors:

The paper is well-structures and well-written.

It can be accepted for publication with minor changes to improve editing.

Author Response

Thank you for your review

Reviewer 2 Report

Hypothesis of this study is, in part, based on seasonal infertility. Therefore, providing a description on the microclimate of Burgos during the study period would be useful; the description “subtropical climate zone” alone is insufficient to describe the local climate. Temperature in a climate zone is influence by distance to sea, ocean current, orientation of mountain ranges and prevailing winds. Reporting the course of the temperature in the housing facility during the study period would be ideal, if data are available. 

Page 3, Table 1: What is “Total born”? If it is litter size, please used the term litter size. If it is different from litter size, please define.   

Page 4, Line 120-121: Do you mean the threshold were higher during the third part (i.e. latter part of the year)? If so, why did you say, “during the summer months”?  Isn’t Burgos in the northern hemisphere? 

P. 5, Line 122-124: Season farrowing rates, an important part of the hypothesis and studyare discussed in these lines, but no seasonal farrowing rates are shown. It would be appropriate to show the seasonal farrowing rates in the table. In addition, when comparing farrowing rates with that in reference 7 and reference 9, please consider that (1) there are likely microclimate differences in the three study sites and (2) the data in the current manuscript is aggregated differently from References 7 and 9. Data based on three-month aggregation or four-month aggregation may be compared and discussed, but needs clarification 

Author Response

All changes are indicated with Track-change.

The reviewer makes valid comments regarding environmental effects. However, we previously failed to mention that the sow barns were environmentally controlled, with cool cells limiting the summer temperatures to a maximum of 28 C. While we can access meteorological (but not barn) data, we do not believe it necessary with the added information.

The litter size description is now improved and line 122-124 has been clarified.

Regarding presentation of farrowing rates, we have not included a breakdown by period. Given the environmental controls and resultant lack of effect of (or interaction with) period means that the details would added little or anything to the paper. We have also clarified the discussion around references 7 and 9.

Reviewer 3 Report

This is an interesting, well written and informative paper which describes an area and trial of high interest to the pig industry (researchers, vets and producers alike). 

There are a couple of relatively minor concerns which should be addressed prior to publication:

1) it might be informative to include the number of piglets born per 100 sows mated for the four treatment groups and during the seasons. 

2) farrowing rates for the different treatments in each seasonal block should be presented in a table or figure

3) need to define whether the superscipts indicate differences between rows or columns in the Table

4) please don't refer to tables in the discussion

Author Response

I personally do not like fecundity index as it is not statistically relevant and if a reader wants it, they can calculate from the a figures in the table. Please see response to reviewer 2. Table legend is now clarified. Table has now been deleted from the discussion.

Please see track-changes in the revised MS